# FUCCI Real-Time Cell-Cycle Imaging as a Guide for Designing Improved Cancer Therapy: A Review of Innovative Strategies to Target Quiescent Chemo-Resistant Cancer Cells

**DOI:** 10.3390/cancers12092655

**Published:** 2020-09-17

**Authors:** Shuya Yano, Hiroshi Tazawa, Shunsuke Kagawa, Toshiyoshi Fujiwara, Robert M. Hoffman

**Affiliations:** 1Department of Gastroenterological Surgery, Okayama University Graduate School of Medicine, Dentistry and Pharmaceutical Sciences, Okayama 700-8558, Japan; htazawa@md.okayama-u.ac.jp (H.T.); skagawa@md.okayama-u.ac.jp (S.K.); toshi_f@md.okayama-u.ac.jp (T.F.); 2Center for Graduate Medical Education, Okayama University Hospital, Okayama 700-8558, Japan; 3Center of Innovative Clinical Medicine, Okayama University Hospital, Okayama 700-8558, Japan; 4Minimally Invasive Therapy Center, Okayama University Hospital, Okayama 700-8558, Japan; 5AntiCancer, Inc., San Diego, CA 92111, USA; all@anticancer.com; 6Department of Surgery, University of California, San Diego, CA 92093, USA

**Keywords:** cell cycle, fluorescent proteins, FUCCI, imaging, targeted cancer therapy, quiescent cancer cells, decoy, chemotherapy

## Abstract

**Simple Summary:**

Chemotherapy of solid tumors has made very slow progress over many decades. A major problem has been that solid tumors very often contain non-dividing cells due to lack of oxygen deep in the tumor and these non-dividing cells resist most currently-used chemotherapy which usually only targets dividing cells. The present review demonstrates how a unique imaging system, FUCCI, which color codes cells depending on whether they are in a dividing or non-dividing phase, is being used to design very novel therapy that targets non-dividing cancer cells which can greatly improve the efficacy of cancer chemotherapy.

**Abstract:**

Progress in chemotherapy of solid cancer has been tragically slow due, in large part, to the chemoresistance of quiescent cancer cells in tumors. The fluorescence ubiquitination cell-cycle indicator (FUCCI) was developed in 2008 by Miyawaki et al., which color-codes the phases of the cell cycle in real-time. FUCCI utilizes genes linked to different color fluorescent reporters that are only expressed in specific phases of the cell cycle and can, thereby, image the phases of the cell cycle in real-time. Intravital real-time FUCCI imaging within tumors has demonstrated that an established tumor comprises a majority of quiescent cancer cells and a minor population of cycling cancer cells located at the tumor surface or in proximity to tumor blood vessels. In contrast to most cycling cancer cells, quiescent cancer cells are resistant to cytotoxic chemotherapy, most of which target cells in S/G_2_/M phases. The quiescent cancer cells can re-enter the cell cycle after surviving treatment, which suggests the reason why most cytotoxic chemotherapy is often ineffective for solid cancers. Thus, quiescent cancer cells are a major impediment to effective cancer therapy. FUCCI imaging can be used to effectively target quiescent cancer cells within tumors. For example, we review how FUCCI imaging can help to identify cell-cycle-specific therapeutics that comprise decoy of quiescent cancer cells from G_1_ phase to cycling phases, trapping the cancer cells in S/G_2_ phase where cancer cells are mostly sensitive to cytotoxic chemotherapy and eradicating the cancer cells with cytotoxic chemotherapy most active against S/G_2_ phase cells. FUCCI can readily image cell-cycle dynamics at the single cell level in real-time in vitro and in vivo. Therefore, visualizing cell cycle dynamics within tumors with FUCCI can provide a guide for many strategies to improve cell-cycle targeting therapy for solid cancers.

## 1. Introduction

In normal cells, the cell cycle is strictly regulated and organized and comprises G_1_-S-G_2_-M phases with an inactive/quiescent G_0_ state. The cell cycle of cancer cells also comprises four distinct phases of the cell cycle [1], but the regulation of the various phases may be altered [2,3,4,5,6]. It is imperative to identify more effective cell-cycle-phase-specific therapeutic agents, especially to target quiescent cells within tumors, which may confer chemoresistance due to their quiescence (Figure 1). The fluorescence ubiquitination cell-cycle indicator (FUCCI) was developed in 2008, which can visualize and identify the cell-cycle phase of cells in real-time [7] (Figure 2). FUCCI utilizes genes linked to different-color fluorescent reporters that are only expressed in specific phases of the cell cycle. FUCCI enables the visualization of the cell cycle of any cell type in real-time. For example, FUCCI imaging enables visualization of real-time cell cycle changes of individual live cancer cells, including during chemotherapy. FUCCI real-time imaging of the cell-cycle phases at the single-cell level in solid tumors in live mice has shown that the vast majority of cancer cells in established experimental tumors are not cycling and remain in the G_0_/G_1_ phase [8,9]. FUCCI imaging showed that cycling cancer cells in the S/G_2_/M phases are located only near the surface of the tumor or near tumor blood vessels [7,8,9]. Cancer cells in the G_0_/G_1_ phase can be the vast majority of cells in solid tumors, which makes cytotoxic chemotherapy of solid tumors often problematic [8,9]. The resistance of quiescent cancer cells to conventional therapy is currently a recalcitrant clinical problem that results in tumor recurrence after chemotherapy and poor prognosis in cancer patients [10,11,12,13,14,15,16]. The present review focuses on the ability of FUCCI to image cell-cycle dynamics within tumors in order to develop a guide for improved targeted therapy of solid tumors, especially against quiescent cancer cells.

## 2. FUCCI Images Cell-Cycle Dynamics of Cancer Cells in Real-Time

To develop cell-cycle targeted cancer therapy, it is critical to monitor cell-cycle phases in real-time for appropriate targeting of therapeutics to specific phases of the cell cycle. As noted above, Sakaue-Sawano et al. [7] developed the FUCCI system for visualizing cell-cycle dynamics of individual cells in real-time by color-coded fluorescence imaging (Figure 2). FUCCI utilizes genes linked to different color fluorescent reporters, such as Cdt1 and Geminin, that only appear in specific phases of the cell cycle. The original FUCCI comprised two plasmids: Cdt1-mKO2 (orange fluorescent protein), expressed in G_1_ phase and Geminin linked to mAG (green fluorescent protein) expressed in late-S-phase. In early-S phase where the expression of Cdt1-mKO2 is the same as that of Geminin-mAG, the cells appear yellow. Therefore, FUCCI-red indicates the quiescent G_1_ phase, FUCCI-green indicates the proliferating late-S/G_2_/M phase, and FUCCI-yellow indicates the early-S phase [7] (Figure 2). Sakaue-Sawano et al. [17] also have developed a new FUCCI with improved color specificity over the original FUCCI, termed FUCCI2. Sakaue-Sawano et al. [18] subsequently developed FUCCI (CA), which produces a sharp triple-color distinct separation of G_1_, S, and G_2_, and can distinguish the transition from G_1_ phase to S phase (Figure 2). The original FUCCI could not distinguish between the G_0_ and G_1_ phases. Therefore, Oki et al. [19] modified FUCCI, which enabled a distinction between G_0_ and G_1_. Bajar et al. [20] created FUCCI4, which distinctly images four cell-cycle phases and enables specific imaging of S phase, with mTurquoise linked with an S-phase-specific protein, stem-loop binding protein (SLBP), and M phase, with mMaroon linked to histone H-1.0 for imaging chromatin condensation during mitosis [20] (Figure 2). FUCCI4 also visualizes the specific cell-cycle phase in real-time, similar to FUCCI (CA). A problem with the original FUCCI was the need to use two separate plasmids, which may not be equally expressed, thereby confusing the color specificity of the various cell-cycle phases. FastFUCCI was subsequently developed, which has an all-in-one expression cassette that has all FUCCI genes under the control of a single promoter, resulting in their equal expression; these different FUCCI systems are compared in Figure 2 [21].

### FUCCI Imaging Demonstrates Different Cell-Cycle Dynamics in 2D- and 3D-Cultured Cancer Cells

FUCCI imaging has visualized the difference in cell-cycle dynamics between two-dimensional (2D) culture, where the majority of cells are in S/G_2_/M, and three-dimensional (3D) culture, where cells are mostly in the G_1_ phase, similar to cancer cells in tumors, as reviewed below [8,9]. This explains, in part, why cancer cells in 2D culture are usually chemosensitive and cancer cells in 3D and in vivo are less chemosensitive or chemoresistant (Figure 3). FUCCI imaging can facilitate the discovery of effective cell-cycle targeting drugs in vitro, especially against quiescent cancer cells, which can predominate in the 3D culture of cancer cells and in vivo.

## 3. Real-time FUCCI Imaging of Cell-Cycle Dynamics of Cancer Cells in Solid Tumors

Intravital in-vivo real-time imaging provides the scene of cancer cells within tumors (Figure 4) [23,24,25,26,27,28,29,30]. Intravital real-time imaging of orthotopic FUCCI-expressing tumors in the liver of live mice was performed at various stages of progression. In these studies, a nascent tumor comprised mostly proliferating cancer cells expressing FUCCI-yellow and FUCCI-green [8,9] (Figure 4). In contrast, an established larger tumor comprised mostly quiescent cancer cells expressing FUCCI-red. In a late-stage tumor, proliferating cancer cells were located only at the surface area or adjacent to blood vessels of tumors (Figure 4) [8,9]. However, it should be noted that same late-stage tumors may have a large number of proliferative cancer cells. FUCCI imaging demonstrated that the cell-cycle phase of each cancer cell in a tumor depends, at least in part, on its location within a tumor; proliferating cancer cells are located near the surface or blood vessels, and quiescent cancer cells are in the central area of tumors, where they become hypoxic and are starving for nutrition [8,9]. Of course, many factors determine the drug resistance of cancer cells, but their location within tumors is important since it can determine access to oxygen and nutrients. Some very well vascularized large tumors may have a greater fraction of cycling cells. Haass et al. [31] and Goto et al. [32] demonstrated that FUCCI-expressing cancer cells were arrested in G_1_ phase during hypoxia and restarted proliferation when oxygen was supplied. Chittajallu et al. [33] also performed intravital FUCCI imaging, in their case, of HT1080 fibrosarcoma cells in a dorsal skin-fold chamber in nude mice and demonstrated that tumors in the chamber comprised approximately 60% of cancer cells in G_1_ phase and approximately 29% of cells in S/G_2_/M phases, similar to the results described above. Haass et al. [31] also demonstrated with FUCCI that quiescent melanoma cells were located far from the tumor vasculature, and proliferating melanoma cells were located near tumor vessels within FUCCI-expressing melanoma tumors [31].

The ability to image the cell cycle of cancer cells within a tumor spatially and temporally can enable the discovery of more effective cell-cycle-targeting therapeutics.

## 4. FUCCI Imaging of Cell-Cycle Phases of Cancer Cells during Chemotherapy and Radiotherapy In Vitro and In Vivo Identifies Resistant Cells

FUCCI imaging of HeLa cancer cells in 2D culture demonstrated that 5-fluorouracil (5-FU) arrests cancer cells in late-S/G_2_ phases without changing the shape of treated cells, and cisplatinum arrests all cancer cells in late-S/G_2_ phases, which acquire a spindle shape. Doxorubicin similarly arrests cancer cells in late-S/G_2_ phases with a spindle shape. Cancer cells surviving doxorubicin were induced to undergo nuclear mis-segregation (Figure 5). FUCCI1 imaging in 2D in-vitro culture demonstrated that chemotherapeutic agents and irradiation arrest some cancer cells in late-S/G_2_/M phases [8,9,21,23]. FUCCI2 imaging of normal murine mammary gland cells (NMuMG) showed that an intermediate concentration of etoposide induced nuclear mis-segregation and a high concentration of etoposide induced DNA endoreplication [17]. Koh et al. [21] used FastFUCCI to show cell-cycle dynamics of cancer cells treated with a taxane. Haass [31] also demonstrated that FUCCI-red-expressing melanoma cells in G_0_/G_1_ phase survived an MEK inhibitor or a BRAF inhibitor. Melanoma spheroids, which survived MEK inhibitors or BRAF inhibitors, recovered and expanded after cessation of treatment, as demonstrated by FUCCI imaging [31]. FUCCI imaging can thus identify the cell cycle phase that drug-resistant cancer cells arrest in, which can suggest subsequent treatment to kill the resistant cells.

FUCCI imaging showed that nascent tumors, comprising mostly cycling cancer cells, are sensitive to chemotherapy. In contrast, quiescent cancer cells within established tumors that survive chemotherapy will restart cycling and increase after the removal of chemotherapeutic agents (Figure 6) [8,9]. Chittajallu et al. [33] also used FUCCI imaging for the quantitative analysis of cancer drug efficacy, spatially and temporally, at the single-cell level in tumors of live mice and demonstrated similar results. Yan et al. [34] demonstrated with FUCCI imaging of rhabdomyosarcoma growing in immunodeficient zebrafish that combination therapy of olaparib, a PARP inhibitor, and temozolomide arrested the rhabdomyosarcoma cells in the G_2_ phase and induced apoptosis.

FUCCI imaging showed that X-irradiated cancer cells were arrested at the G_2_ checkpoint in vitro [35]. Onozato et al. [35] showed that irradiation decreased FUCCI-green-expressing proliferative cancer cells and increased FUCCI-red-expressing quiescent cancer cells in vitro. Bouchard et al. [36] also demonstrated that there are more FUCCI-red-expressing quiescent cancer cells in irradiated mammary glands than control mammary glands.

## 5. FUCCI Imaging Identified Angiogenesis Occurring Among Resistant Cells During Chemotherapy of Tumors

Angiogenesis and anti-angiogenesis therapeutics are still active areas of cancer biology research [37,38]. A transgenic nude-mouse model, in which the nestin promoter drives GFP expression and labels nascent blood vessels with GFP within tumors [39] (Figure 7), was used to visualize angiogenesis, along with FUCCI imaging, which demonstrated that cycling cancer cells are located near tumor vessels and quiescent cancer cells are located far from tumor vessels (Figure 7) [8,9]. FUCCI imaging with GFP-expressing tumor vessels showed nascent tumor blood vessels and increased angiogenesis occurring among the chemo-resistant cancer cells within the tumor. Angiogenesis increased within areas where quiescent resistant cancer cells survived after chemotherapy (Figure 8) [40]. This is another reason why quiescent cancer cells surviving chemotherapy can rapidly restart cycling after cessation of chemotherapy and become more aggressive. Tumors can vary with regard to their degree of vascularization, which must be taken into consideration. Ironically, cancer cells that become resistant to cytotoxic agents not only become much more refractory to chemotherapy but can become more aggressive and more metastatic [41,42]. FUCCI imaging can provide a very useful tool to discover drugs that can overcome the chemoresistance of cancer cells.

## 6. Use of FUCCI to Identify and Target Quiescent Cancer Cells In Vitro and In Vivo

Quiescent cancer cells are important therapeutic targets as they can resist most cytotoxic chemotherapy, then cycle after the cessation of therapy. This section focuses on novel strategies afforded by FUCCI imaging to identify, decoy, attack, and kill quiescent cancer cells in tumors, as well as in vitro.

### 6.1. FUCCI Imaging Can Identify and Target Quiescent Cancer Stem Cells

Cancer stem cells (CSCs) have the ability to self-renew and initiate tumors [43,44]. CSCs are resistant to cytotoxic agents since they have more drug transporters, are more protective against reactive oxygen species (ROS), and proliferate more slowly than non-CSCs. Previously, before FUCCI was developed, it was difficult to monitor the cell-cycle phase dynamics of CSCs. FUCCI imaging showed that CSCs are quiescent compared with non-CSCs, even in two-dimensional culture [45]. Spheroid 3D culture is frequently used to grow CSCs. FUCCI imaging demonstrated that spheroids derived from CSCs are arrested in the G_1_ phase for more than 8 days [45]. Using FUCCI real-time imaging, CSCs were imaged to begin to divide when fetal bovine serum (FBS) was added to the culture medium [45]. Haass et al. [31] also demonstrated that FUCCI-expressing melanoma spheroids comprised FUCCI-red quiescent cells. In contrast, FUCCI melanoma spheroids started proliferation and cell division when FBS was added. FUCCI imaging can facilitate the discovery of effective cell-cycle targeting drugs specific for quiescent CSCs, as discussed below.

### 6.2. Decoy of Quiescent Cancer Stem Cells to Commence Cycling with a Tumor-Targeting Adenovirus, Demonstrated by FUCCI Imaging

FUCCI imaging showed that quiescent CSCs were resistant to conventional chemotherapy, as mentioned above [45]. Therefore, a cell-cycle decoy of CSCs to cycle would be a promising therapeutic option. Using a genetically-engineered telomerase-specific tumor-targeting adenovirus, OBP-301 [46,47], FUCCI imaging showed that the adenovirus decoyed and trapped quiescent CSCs from the G_1_ phase into S/G_2_ phases, where the decoyed CSCs became sensitive to chemotherapy in vitro and in vivo (Figure 8) [45].

### 6.3. Tumor-Targeting Salmonella Typhimurium Decoys Quiescent Cancer Cells in Tumors to Cycle and Become Chemosensitive, Shown by FUCCI Imaging

FUCCI imaging showed that tumor-targeting *Salmonella typhimurium* (*S. typhimurium*) A1-R [49] decoyed quiescent cancer cells in G_0_/G_1_ phase cultured in monolayer culture, as tumor spheres in three-dimensional culture, and tumors in vivo, to begin cycling (Figure 9) [50]. *S. typhimurium* A1-R, combined with chemotherapy, inhibited tumor growth compared with *S. typhimurium* A1-R monotherapy or chemotherapy alone [50]. FUCCI imaging demonstrated that the decoyed tumor comprised mostly cancer cells in S/G_2_M phases, which became sensitive to chemotherapy. The cell-cycle decoy ability of *S. typhimurium* A1-R, developed with FUCCI imaging, can lead to a new paradigm of targeting quiescent cancer cells.

### 6.4. Decoy, Trap, and Kill Cancer Therapy Developed with FUCCI Imaging

FUCCI imaging showed that a tumor-targeting adenovirus decoyed and trapped both quiescent CSCs and quiescent established tumors from G_1_ phase to early-S phase, as mentioned above. The CSCs in early-S phase, decoyed and trapped by the adenovirus, became sensitive to chemotherapy [45] (Figure 9). *S. typhimurium* A1-R also decoyed quiescent cancer cells in solid tumors to cycle. After the decoy, the cancer cells were trapped in S/G_2_ under methionine restriction effected by recombinant methioninase (rMETase). The cell-cycle trap of cancer cells by methionine restriction was clearly shown by FUCCI imaging [51]. The tumors decoyed by *S. typhimurium* and trapped by rMETase became significantly sensitive to conventional cytotoxic agents [52] (Figure 10). This novel treatment strategy has been termed decoy, trap, and kill chemotherapy.

### 6.5. FUCCI Imaging Demonstrates that Invading Cancer Cells Are Quiescent

FUCCI imaging showed that as cancer cells began to invade, they become quiescent in G_0_/G_1_, expressing FUCCI-red [24]. FUCCI imaging showed that as cancer cells cultured on collagen began to invade, they also become quiescent in G_0_/G_1_, expressing FUCCI-red [53]. Miyashita et al. [54] showed that the leading edge of invading cells expressed FUCCI-red cancer cells, indicating they were in G_1_ phase. These results further show the danger of quiescent cancer cells and the importance of targeting them.

## 7. FUCCI Imaging to Evaluate Cell-Cycle-Specific Targeted Drugs

CDK4 and CDK6 play a role in malignant progression [55,56,57,58,59,60,61,62] (Figure 1). FUCCI imaging demonstrated that the response of FUCCI-expressing HeLa cells to a Cdk4 inhibitor depended on the cell-cycle phase. Exposure to a Cdk4 inhibitor in early G_1_, G_1_/S, or S/G_2_ phases resulted in G_1_ arrest. In contrast, exposure to a Cdk4 inhibitor in late-G_1_ phase resulted in the endoreplication of the cellular DNA in the G_1_ phase (17). These results indicate that a CDK inhibitor [63,64,65,66,67,68,69,70,71] induces G_1_ arrest. WEE 1 kinase is essential to bypass the G_2_ checkpoint [72]. WEE1 inhibitors are known to induce abortive mitosis (endomitosis) since cancer cells have polypoid nuclei after WEE inhibition [72]. Nojima et al. [73] demonstrated using FUCCI imaging that mitosis of cancer cells was extended when the WEE1 inhibitor was applied, and surviving cells were in G_1_ and expressed FUCCI-red with endomitosis. The present review demonstrates that FUCCI can be a useful tool to evaluate and discover cell-cycle-targeting drugs [74,75,76] and improve the efficacy of currently-used cell-cycle-targeting drugs.

## 8. Comparison of FUCCI and Other Cell-Cycle Indicators

There are other methods to monitor the cell cycle. Many types of functional nuclear proteins, such as histone H2B [77], nuclear localization signal (NLS) [78,79], and proliferating-cell nuclear antigen (PNCA) [80,81], have been used to monitor specific cell-cycle phases with fluorescent proteins. Cancer cells transfected with the histone H2B gene fused to the gene-encoding GFP have been used to monitor mitotic cells in real-time [82]. Dual-fluorescence-colored cancer cells with histone H2B-GFP and cytoplasm-RFP have also been used to monitor mitosis and nuclear–cytoplasmic interaction in real-time [83,84,85,86,87]. However, this system only distinguishes mitotic and nonmitotic cells. Recently, Sparks et al. [88] demonstrated doxorubicin-induced chromatin-changes in-vivo using histone-GFP and a fluorescence lifetime imaging microscope (FLIM) [89]. This system is an improvement over the dual-color fluorescent cells and gives more information about mitosis but cannot indicate all cell cycle phases as FUCCI can [89]. Fusing NLS to fluorescent proteins enabled visualization of cell-cycle transition from G1 phase to S phase in real-time [78,79]. PCNA linked to fluorescent proteins enabled visualization of cell-cycle transition from S to G2 phase in real-time [89,90]. However, these makers cannot track the cell-cycle dynamics of all phases, as modern FUCCI can. Foster resonance energy transfer (FRET)-image sensors have been used with CDKs-CFP and YFP to monitor cell cycle transition, but cannot visualize all cell-cycle phases, as modern FUCCI can [91,92,93,94]. FUCCI is the most useful tool to monitor all phases of cell-cycle dynamics in real time [95,96].

## 9. Conclusions

The resistance of quiescent cancer cells to conventional therapy is currently a recalcitrant clinical problem. FUCCI imaging demonstrated that most currently-used cytotoxic cancer chemotherapeutic agents are effective only on cycling cancer cells and have little effect on quiescent/dormant cancer cells that comprise the majority of the cells within many, if not most, established tumors. FUCCI imaging can be used to discover drugs that target quiescent cancer cells, a mainly unsolved problem in oncology.

FUCCI imaging in real-time has demonstrated that the cell-cycle position of each cancer cell in a solid tumor is a major determinant of whether it can respond to cytotoxic chemotherapy. A very high proportion of G_0_/G_1_ quiescent cells in solid tumors makes the tumor resistant to cytotoxic chemotherapy. FUCCI imaging has demonstrated that the spatial location of cancer cells within a tumor has a major impact on their cell-cycle behavior. FUCCI imaging can be used to design new cell-cycle targeted chemotherapy, including using cell-cycle decoy agents to overcome this recalcitrant problem of chemotherapy resistance of quiescent cells in tumors (Figure 11). Modern FUCCI imaging has critical advantages over all other current cell-cycle analyses in that it can image and distinguish all phases of the cell cycle in real-time. FUCCI can be a powerful tool for the discovery of paradigm-changing cancer therapy in the clinic, such as decoy, trap, and kill therapy, which can have applications in the near future.

## Figures and Tables

**Figure 1 cancers-12-02655-f001:**
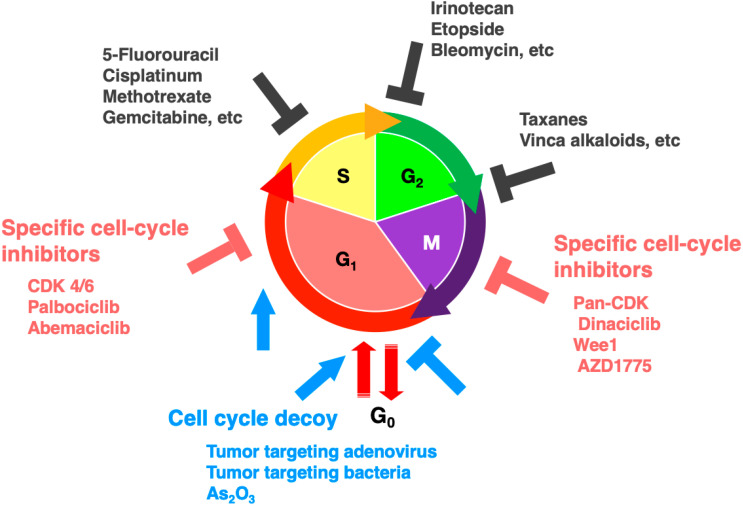
Scheme of cell-cycle phases where therapeutic agents are most effective, which can be imaged with the fluorescence ubiquitination cell-cycle indicator (FUCCI).

**Figure 2 cancers-12-02655-f002:**
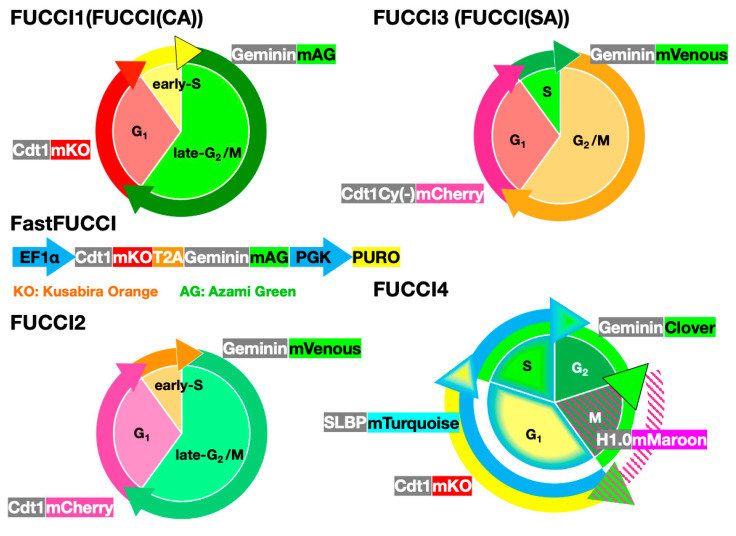
Comparison of the different FUCCI systems.

**Figure 3 cancers-12-02655-f003:**
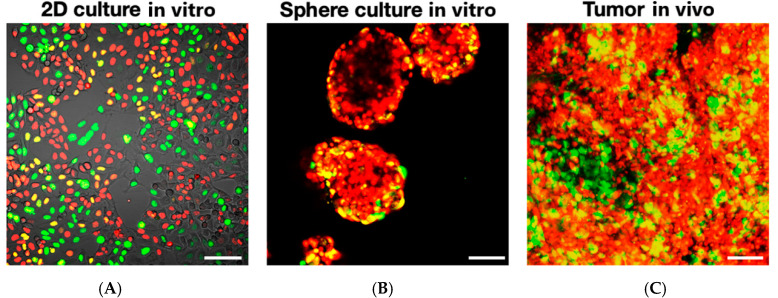
Representative FUCCI image of MKN45 cancer cells in 2D and 3D culture and in vivo. FUCCI cells in 2D in vitro (**A**). FUCCI cells in 3D sphere culture in vitro (**B**). FUCCI imaging of cancer cells in tumors in vivo (**C**). The cells in G_1_, early-S, or late-S/G_2_/M phases appear in red, yellow, or green, respectively. Note: the preponderance of FUCCI-green proliferating cells in 2D-culture and the preponderance of FUCCI-red quiescent cells in 3D-culture and in tumors. [22]. Scale bar; 100 μm.

**Figure 4 cancers-12-02655-f004:**
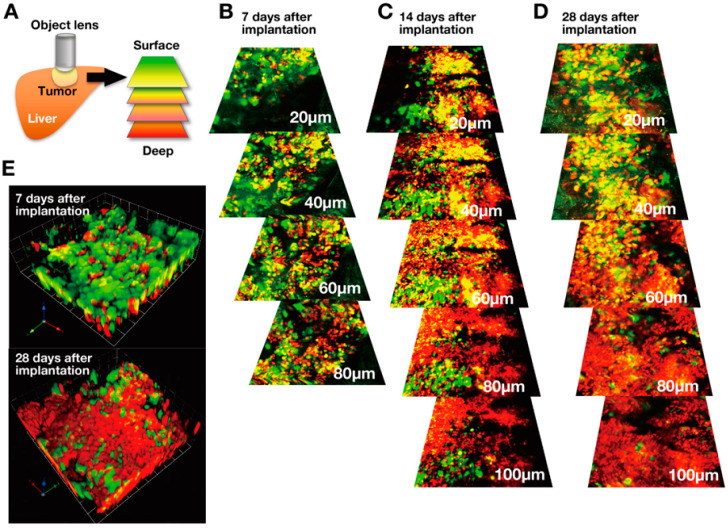
Intravital FUCCI imaging of tumors at various stages and depths. (**A**) Scheme of intravital imaging of FUCCI cancer cells in the liver of nude mice with a confocal laser microscope. (**B**–**D**) Representative images of FUCCI-expressing tumors at various stages, at the indicated depth from the surface. (**E**) Representative images of 3D reconstruction of FUCCI tumors at 7 and 28 days of growth. The cells in G_1_, early-S, or late-S/G_2_/M phases appear red, yellow, or green, respectively [8].

**Figure 5 cancers-12-02655-f005:**
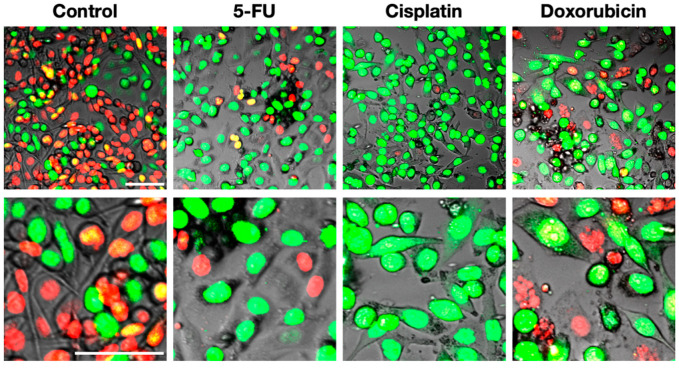
Representative images of responses of FUCCI HeLa cells to cytotoxic agents. FUCCI HeLa cancer cells were treated with 5-FU, cisplatinum, and doxorubicin. The cells in G_1_, early-S, or late-S/G_2_/M phases appear red, yellow, or green, respectively [32]. Scale bar; 100 μm.

**Figure 6 cancers-12-02655-f006:**
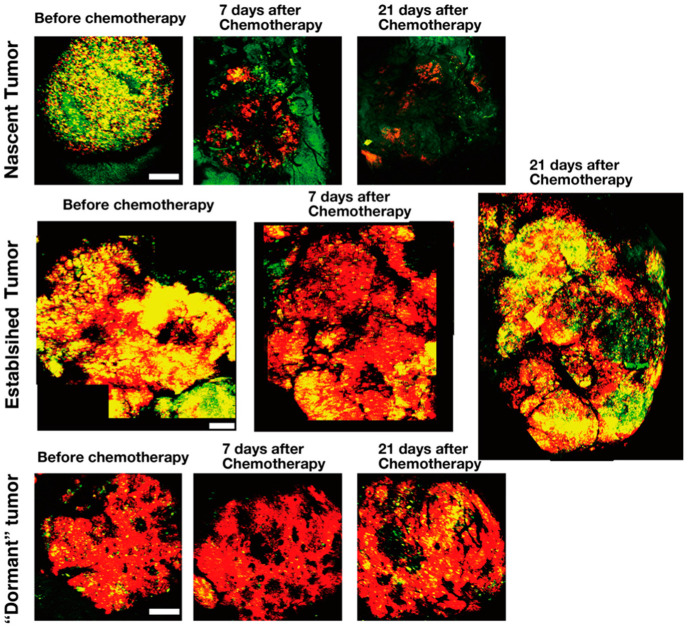
Response to cytotoxic chemotherapy of tumors at different stages, visualized by FUCCI. Representative images of FUCCI-expressing tumors treated with cisplatinum. The cells in G_0_/G_1_, early S, or late S/G_2_/M phases appear red, yellow, or green, respectively [8]. Scale bar; 500 μm.

**Figure 7 cancers-12-02655-f007:**
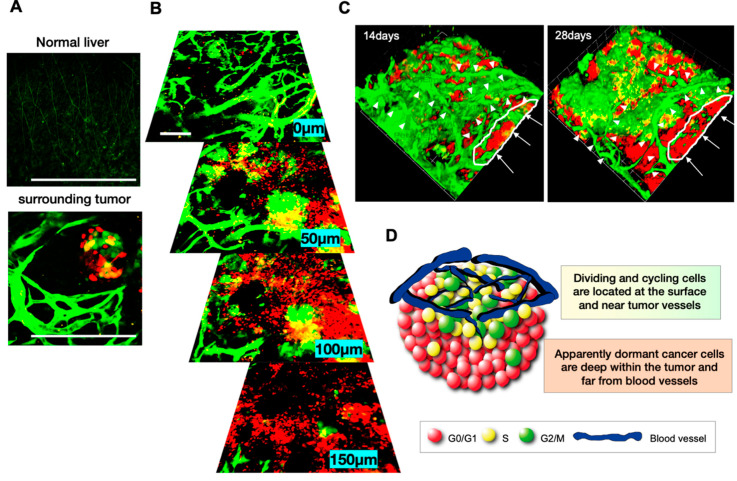
Intravital imaging of GFP-expressing tumor blood vessels and FUCCI-expressing cancer cells in tumors at different stages and at depths within tumors. (**A**) Representative images of tumor angiogenesis in nude mice expressing nestin-driven GFP in nascent blood vessels. Cancer cells induce tumor angiogenesis. Scale bar; 500 μm. (**B**) Representative images of FUCCI-expressing tumors with GFP-expressing tumor vessels at the indicated depth from the surface. (**C**) Representative images of 3D reconstruction of FUCCI tumors, with visualized tumor vessels at 14 days or 28 days after implantation. Arrowheads indicate tumor vessels. Arrows show quiescent areas of the tumor. (**D**) Scheme of FUCCI cancer cells and tumor vessels [8,22].

**Figure 8 cancers-12-02655-f008:**
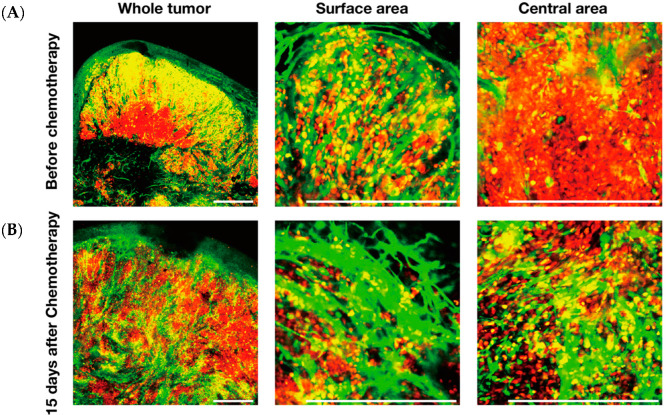
Cytotoxic chemotherapy induces tumor angiogenesis. (**A**) Representative images of FUCCI-expressing tumors with visualized GFP-expressing tumor vessels before chemotherapy. (**B**) Representative images of FUCCI-expressing tumors with visualized GFP-expressing tumor vessels after chemotherapy. Elongated green structures are GFP-expressing nascent tumor vessels formed in transgenic mice expressing nestin-driven GFP [40,48]. Scale bar; 500 μm.

**Figure 9 cancers-12-02655-f009:**
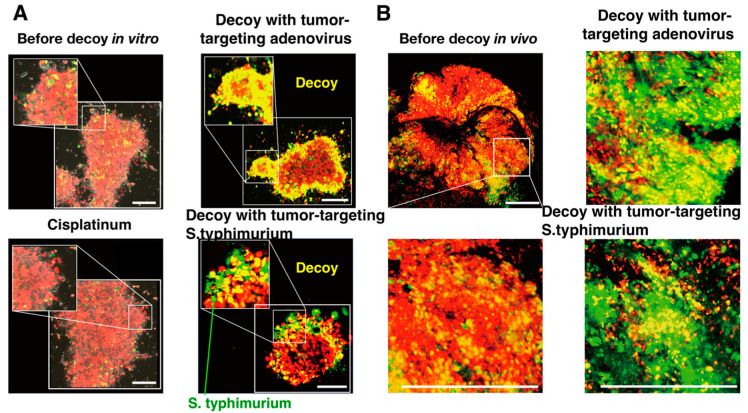
Cell-cycle decoy of tumor-targeting adenovirus and tumor-targeting *S**almonella typhimurium* A1-R, observed with FUCCI imaging. (**A**) Representative images of the decoy of quiescent cancer cells in vitro, before and after decoy. Tumor-targeting adenovirus and tumor targeting *S. typhimurium* A1-R decoy quiescent cancer cells in tumor spheres from G1 into early-S and late-S/G_2_ phases (right). (**B**) Representative images of decoy of quiescent cancer cells in vivo. Tumor-targeting adenovirus and *S. typhimurium* decoy quiescent cancer cells in tumors in vivo into early-S and late-S/G_2_ phases [22]. Scale bar; 500μm.

**Figure 10 cancers-12-02655-f010:**
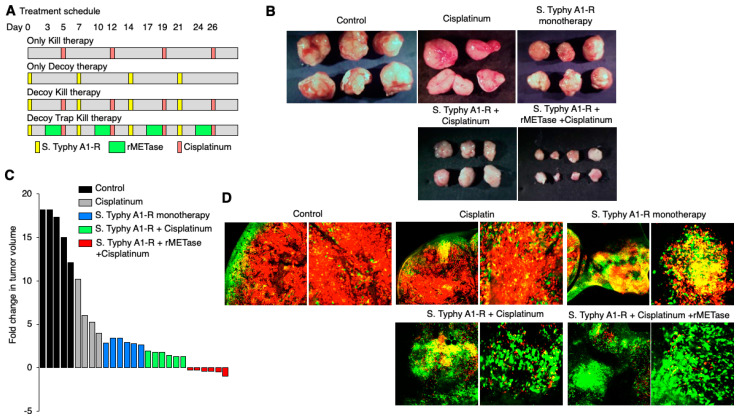
Decoy, trap, and kill chemotherapy with FUCCI imaging. FUCCI-expressing MKN45 stomach cancer cells (5 × 10^6^ cells/mouse) were injected subcutaneously into the left flank of nude mouse. When the tumors reached approximately 8 mm in diameter (tumor volume, 300 mm^3^), mice were administered iv *S. typhimurium* A1-R alone or with cisplatinum (CDDP; 5 mg/kg ip) for 5 cycles every 7 days, in combination with *S. typhimurium* A1-R and CDDP or in combination with *S. typhimurium* A1-R, recombinant methioninase (rMETase; dose ip 200 u/day 3 days/cycle), and CDDP (5 mg/kg, ip). (**A**) Treatment schedule. (**B**) Macroscopic photographs of FUCCI-expressing MKN45 subcutaneous tumors, untreated control: *S. typhimurium* A1-R-treated, CDDP-treated, or treated with the combination of *S. typhimurium* A1-R and CDDP or the combination of *S. typhimurium* A1-R, rMETase, and CDDP. (**C**) Waterfall plot indicating fold change in tumor volume with each treatment. (**D**) Representative images of cross-sections of FUCCI-expressing MKN45 subcutaneous tumors, untreated control: *S. typhimurium* A1-R-treated, CDDP-treated, or treated with the combination of *S. typhimurium* A1-R and CDDP or the combination of *S. typhimurium* A1-R, rMETase, and CDDP [45]. iv = intravenous; ip = intraperitoneal. Scale bar; 500 μm.

**Figure 11 cancers-12-02655-f011:**
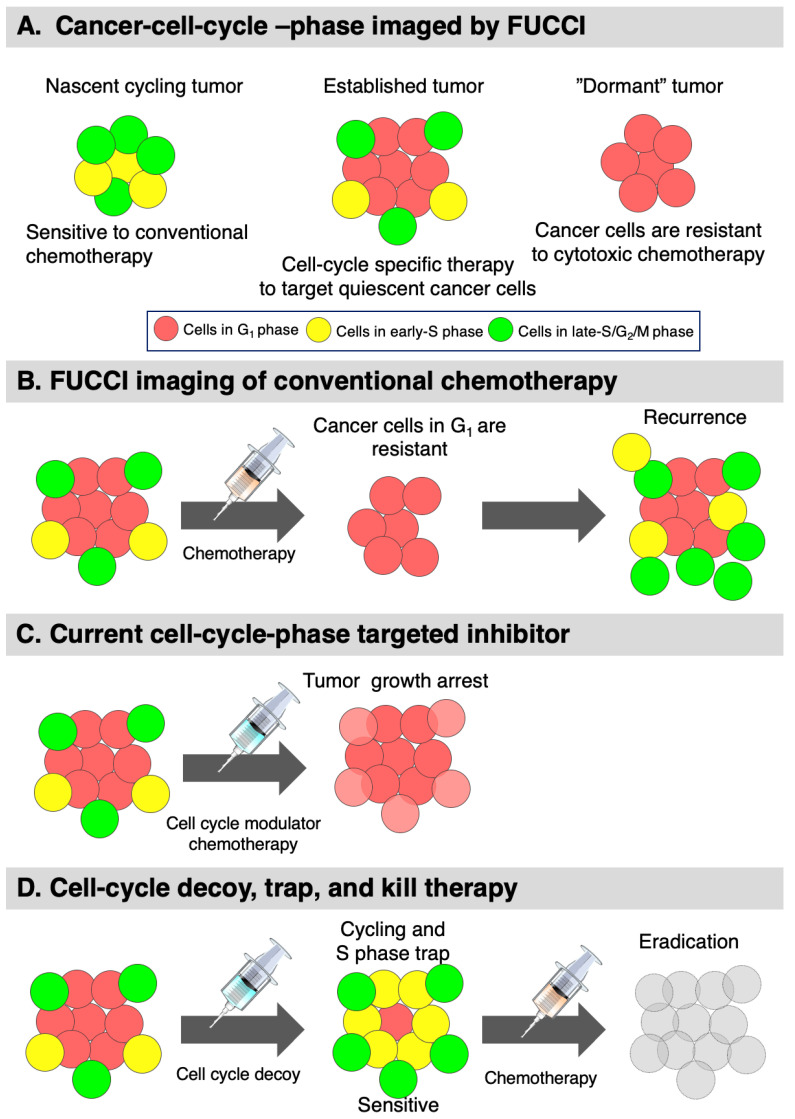
FUCCI image-guided cell-cycle targeted therapy. (**A**) FUCCI imaging of solid tumors at different stages: a nascent tumor comprising mostly cycling cancer cells, an established tumor consisting of mostly quiescent cancer cells, and some cycling cancer cells. A dormant tumor consisting of mostly quiescent cancer cells. (**B**) FUCCI imaging of chemotherapy of an established tumor. (**C**) FUCCI-guided cell-cycle-specific targeting with current inhibitors. (**D**) Cell-cycle decoy, trap, and kill chemotherapy.

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
