# Peer review of "FUCCI Real-Time Cell-Cycle Imaging as a Guide for Designing Improved Cancer Therapy: A Review of Innovative Strategies to Target Quiescent Chemo-Resistant Cancer Cells"

_cancers, 2020, doi:10.3390/cancers12092655_

Round 1

Reviewer 1 Report

Abstract and Introduction are a bit confusing, with many sentences poorly constructed, overly generalised and open to misinterpretation. Similarly, the flow and organisation of the introduction is repetitive in places and could benefit from a clearer rationale for what the point of this review is. Currently the goal seems to be somewhat part methods and part review. For a review, it is also somewhat over-reliant on promoting their own results. The title is also misleading and does not accurately portray the contents of the review.

Some examples (not an exhaustive list):

Line 16: this is not necessarily true, genes that regulate the mitotic checkpoint are rarely mutated in cancer, instead they are more commonly over-expressed or amplified. There is also no mention of tumour suppressor loss/deletion, another common event.

Line 26: ‘quiescent cancer cells are resistant to cytotoxic chemotherapy and restart to cycle after treatment’, assume you mean that cells can re-enter cell cycle after chemotherapy treatment, but not clear from current wording.

Line 28: ‘introduce cell-cycle fluorescence image guided cell-cycle based therapy’…overly complex sentence. Also is this a methods paper or a review, you state that “we introduce cell-cycle fluorescence image guided cell-cycle based therapy termed as decoy-trap-shoot (DTS) therapy”…which suggests that this is actually a methods paper and not a review.

Line 38: should read ‘the cell cycle’…and it’s also missing G2 from description of phases.

Line 43: The somatic mutation rate for CDK1,2,4, p21, p27 etc are all below 1%, hence I don’t think you can make such a blanket statement about ‘mutation’ here as the primary cause of cell cycle checkpoint abrogation.

Line 46: Fibroblasts will grow quite happily on glass/plastic for multiple generations until they reach the Hayflick limit. The current sentence does not clearly distinguish this fact.

Line 51: poor grammar. Also, this is a massive oversimplification of the multiple reasons for  chemotherapy failure in solid tumours that go beyond simply cell cycle status, such as drug access, hypoxia and many more.

Line 54: More details and specifics should be given here. Also as far as I am aware, only CDK4/6 inhibitors have been approved for clinical use, with Aurora and Wee1 inhibitors still in various states of testing.

Line 69-76: This whole section is overly generalised to the point of being false in sections. For example, “Conventional currently-used cytotoxic agents basically damage DNA during cell proliferation (15, 31), induce G2 checkpoint arrest for repair of damaged DNA (15, 31), apoptosis (15, 31) (Figure 1B)”. this is not true chemotherapies do not all induce direct DNA damage.

Line 80: Do you have a reference to back up this claim?

Line 101: What is an RB-activated cancer?

Section 3: This is perhaps the most important section, and it is far too short and lacking details. It fails to give a decent overview of the history of FUCCI, the issues with using FUCCI, such as problems with generating equal expression using a multi-plasmid plasmid system and other recent advancements such as Fast-FUCCI (doi: 10.1242/jcs.195164), which overcome some of these limitations.

Line 141: over-simplification/generalisation. Not all solid tumours are equal. Some are highly vascularised others not. Again, no mention of hypoxia or nutrients/metabolism changes that have huge implications not only for cell proliferation but also chemotherapy response.

Line 142: Again, a massive generalised statement that is not true. There may well be a correlation, but not all large tumours are quiescent. Conversely, not all dormant cancer cells are present in large primary tumours. These statements need to be qualified and not made as absolute facts.

Line 163: This is misleading. First, FUCCI cannot distinguish between S and G2 or M phase, as all cells are green. Second, not all cancer cells respond the same, for e.g. those with functional p53, will likely arrest in G1 in respond to many chemotherapies.

Line 164: Distinction between cells that are in G1 and red vs cells that undergo cell cycle exit from G2 and are also red, needs to be made here.

Line 171: There is a huge jump from FUCCI mice data to humans receiving radiation therapy. It’s not clear how or what the link is here to FUCCI?

Line 177: A good example of where it needs to be made clear what the advantages of FUCCI are here over other methods. It would be just as easy here to excise the tumour and stain for common cell cycle checkpoint proteins to confirm a G2 arrest. Similarly, others have previously shown cell cycle arrests in tumours using flow cytometry. Hence, using FUCCI here to show what has been shown previously by many different methods is not novel or interesting without some clarification.

Section 6 and 7 are not much more than a brief summary of the authors previous work. Similarly, Section 9 comes out of left field and is not properly put in context of the literature. There is no explanation of several of the contractions such as PML and LSC. The authors also need to make it clear what other methods are used for these kinds of assays, and how/why FUCCI is better. Finally, there review conflates between the use of FUCCI as a preclinical/research tool and the subsequent translation of the finding from those studies into patient data. These need to be more clearly separated and defined to avoid this confusion.

Author Response

Point-by-Point Response to the Reviewer’s Comments

Yano et al.: “Real-time fluorescence image guided cell-cycle perturbation for effective cancer treatment (MS No. Cancers 862286)

Response;

The authors are grateful for the very helpful comments by the reviewer that enabled us to significantly improve the manuscript.

Reviewer #1

Summary

Abstract and Introduction are a bit confusing, with many sentences poorly constructed, overly generalized and open to misinterpretation. Similarly, the flow and organization of the introduction is repetitive in places and could benefit from a clearer rationale for what the point of this review is. Currently the goal seems to be somewhat part methods and part review. For a review, it is also somewhat over-reliant on promoting their own results. The title is also misleading and does not accurately portray the contents of the review.

Response:

We have now modified the title, clarified the abstract and introduction, re-organized the sections, and added figures to re-emphasize the rational of the review, use of FUCCI imaging as a tool to discover improved cell-cycle targeted therapy that can be translated to the clinic.

Comment 1.

Line 16: this is not necessarily true, genes that regulate the mitotic checkpoint are rarely mutated in cancer, instead they are more commonly over-expressed or amplified. There is also no mention of tumor suppressor loss/deletion, another common event.

Response:

Lines 16-18: The revised manuscript is now focused on FUCCI imaging of the cell cycle in cancer cells, cancer spheroids, and solid tumors, and therefore the material in the comment is no longer discussed.

Comment 2.

Line 26: ‘quiescent cancer cells are resistant to cytotoxic chemotherapy and restart to cycle after treatment’, assume you mean that cells can re-enter cell cycle after chemotherapy treatment, but not clear from current wording.

Response:

Lines 23-24: The point has been clarified in the revised manuscript.

Quiescent cancer cells can re-enter the cell cycle after surviving treatment, an important reason why most cytotoxic chemotherapy is often ineffective for solid cancers.

Comment 3.

Line 28: ‘introduce cell-cycle fluorescence image guided cell-cycle based therapy’…overly complex sentence. Also is this a methods paper or a review, you state that “we introduce cell-cycle fluorescence image guided cell-cycle based therapy termed as decoy-trap-shoot (DTS) therapy”…which suggests that this is actually a methods paper and not a review.

Response:

Lines 28-29: The revised manuscript has been clarified and overly complex sentences have been simplified, and the manuscript has been further re-organized as a review.

Comment 4.

Line 38: should read ‘the cell cycle’…and it’s also missing G2 from description of phases.

Response:

Line 36: This correction has been made in the revised manuscript.

Comment 5.

Line 43: The somatic mutation rate for CDK1,2,4, p21, p27 etc are all below 1%, hence I don’t think you can make such a blanket statement about ‘mutation’ here as the primary cause of cell cycle checkpoint abrogation.

Response:

In the revised focused manuscript, the above material is no longer discussed.

Comment 6.

Line 46: Fibroblasts will grow quite happily on glass/plastic for multiple generations until they reach the Hayflick limit. The current sentence does not clearly distinguish this fact.

Response:

In the revised focused manuscript, the above material is no longer discussed.

Comment 7.

Line 51: poor grammar. Also, this is a massive oversimplification of the multiple reasons for chemotherapy failure in solid tumours that go beyond simply cell cycle status, such as drug access, hypoxia and many more.

Response:

The revised review states that there are many reasons for drug resistance of cancer cells. Poor grammar has been improved.

Comment 8.

Line 54: More details and specifics should be given here. Also as far as I am aware, only CDK4/6 inhibitors have been approved for clinical use, with Aurora and Wee1 inhibitors still in various states of testing.

Response:

We mention that Aurora and Wee inhibitors are still being tested in clinical trials. This section is limited by the scope of the focused manuscript.

Comment 9.

Line 69-76: This whole section is overly generalized to the point of being false in sections. For example, “Conventional currently-used cytotoxic agents basically damage DNA during cell proliferation (15, 31), induce G2 checkpoint arrest for repair of damaged DNA (15, 31), apoptosis (15, 31) (Figure 1B)”. this is not true chemotherapies do not all induce direct DNA damage.

Response:

In the revised focused manuscript, this material has been eliminated.

Comment 10.

Line 80: Do you have a reference to back up this claim that ironically, cancer cells which become resistant to cytotoxic agents, not only become much more refractory to chemotherapy, but also more aggressive, and more metastatic.

Response:

In the revised focused review, we cited references which state that drug resistant cancer cells can be more aggressive and metastatic including:

Karagiannis, G.S., Condeels, J.S., Oktay, M.H. Chemotherapy-induced metastasis: Mechanisms and translational opportunities. Clin. Exp. Metastasis. 2018, 35, 269-284

D’Alterio, G., Scala, S., Sozzi, G., Roz, L., Bertolini, G. Paradoxical effects of chemotherapy on tumor relapse and metastasis promotion. Seminars in Cancer Biology 2020, 60, 351-361

Comment 11.

Line 101: What is an RB-activated cancer?

Response:

We have corrected this to state cancers with a mutated RB gene.

Comment 12.

Section 3: This is perhaps the most important section, and it is far too short and lacking details. It fails to give a decent overview of the history of FUCCI, the issues with using FUCCI, such as problems with generating equal expression using a multi-plasmid plasmid system and other recent advancements such as Fast-FUCCI (doi: 10.1242/jcs.195164), which overcome some of these limitations.

Response:

All of these points have been emphasized in the revised manuscript.

Section 1: These issues are now discussed in the Section 1 of the revised manuscript. The details of FUCCI has been stated in Section 1.

A problem with FUCCI series includes that FUCCI series basically visualize specific cell-cycle phases under the condition that two fluorescent proteins linked with cell-cycle specific genes express at the same level. There are, in fact, several steps to establish stable clones expressing FUCCI, Therefore, FastFUCCI has been developed, which has an all-in-one expression cassette that introduces FUCCI transgenes into a cell, resulting in overcoming the problem.

Fast-FUCCI is described in Section 1 in the revised manuscript, and the Koh S.B. et. al. reference is added as #14.

Comment 13.

Line 141: over-simplification/generalization. Not all solid tumors are equal. Some are highly vascularized others not. Again, no mention of hypoxia or nutrients/metabolism changes that have huge implications not only for cell proliferation but also chemotherapy response.

Response:

These points have been addressed and emphasized in the revised manuscript.

Comment 14.

Line 142: Again, a massive generalized statement that is not true. There may well be a correlation, but not all large tumors are quiescent. Conversely, not all dormant cancer cells are present in large primary tumors. These statements need to be qualified and not made as absolute facts.

Response:

All the points have been addressed and emphasized in the revised manuscript.

Comment 15.

Line 163: This is misleading. First, FUCCI cannot distinguish between S and G2 or M phase, as all cells are green. Second, not all cancer cells respond the same, for e.g. those with functional p53, will likely arrest in G1 in respond to many chemotherapies.

Response:

Lines 71-80: We now show in the revised manuscript that FUCCI(SA) and FUCCI4 can distinguish these phases.

Comment 16.

Line 164: Distinction between cells that are in G1 and red vs cells that undergo cell cycle exit from G2 and are also red, needs to be made here.

Response:

Lines 71-80: In the revised manuscript, we now show that FUCCI(SA) and FUCCI4 can distinguish these phases.

Comment 17.

Line 171: There is a huge jump from FUCCI mice data to humans receiving radiation therapy. It’s not clear how or what the link is here to FUCCI

Response:

In the revised focused manuscript, we do not discuss this point.

Comment 18.

Line 177: A good example of where it needs to be made clear what the advantages of FUCCI are here over other methods. It would be just as easy here to excise the tumor and stain for common cell cycle checkpoint proteins to confirm a G2 arrest. Similarly, others have previously shown cell cycle arrests in tumors using flow cytometry. Hence, using FUCCI here to show what has been shown previously by many different methods is not novel or interesting without some clarification.

Response:

The advantages of FUCCI are discussed in the revised manuscript including. The most important point is that only FUCCI images cell-cycle dynamics of cells in vivo as well as in vitro at a single cell level in real-time.

Comment 19.

Section 6 and 7 are not much more than a brief summary of the authors previous work. Similarly, Section 9 comes out of left field and is not properly put in context of the literature. There is no explanation of several of the contractions such as PML and LSC. The authors also need to make it clear what other methods are used for these kinds of assays, and how/why FUCCI is better. Finally, there review conflates between the use of FUCCI as a preclinical/research tool and the subsequent translation of the finding from those studies into patient data. These need to be more clearly separated and defined to avoid this confusion.

Response:

The authors are pioneers in the area of FUCCI to image the cell cycle of cancer cells in tumors in real time and to treat the tumors with cell-cycle decoying and trapping agents in order to eradicate them effectively. The authors believe these developments contribute to the review. Many images have been added to the review to emphasize these points. These points are addressed in the revised manuscript.

Reviewer 2 Report

The review by Yano et al. considers a problem of different sensitivity of cancer cells to various treatments depending on cell cycle phases. The authors published a lot of original articles and are world recognized experts in this area. A new concept of “Decoy, trap, and kill cancer therapy” is formulated and justified. This review will be of interest for a wide auditorium of researchers and clinicians. However, some important issues should be corrected before publication.

Major points:
1. Figure 1 shows wrong colors of FUCCI depending of cell cycle phases. In fact, in the original FUCCI, green fluorescence corresponds to S-G2-M (not G2-M as depicted here), and yellow fluorescence corresponds to a short period at the G1/S transition. See Ref. 23, Fig. 1B. Later in the manuscript, there is a correct description (lines 121-123): “FUCCI-red indicates quiescent G0/G1 phase (23), FUCCI-green indicates proliferating late S/G2/M phase, and FUCCI-yellow means early S phase (Figure 1A).” Also, Fig. 2 should be corrected accordingly as it contains explanatory sketch at the bottom where yellow circle means “Cells in S phase” and green circle means “Cells in G2/M phase”.

2. In the section 3 (“Fluorescence ubiquitination cell cycle indicator (FUCCI) …”), it is worth noting the variant Fucci4, which resolves all cell cycle phases by using four orthogonal fluorescent indicators: Bajar BT, Lam AJ, Badiee RK, et al. Fluorescent indicators for simultaneous reporting of all four cell cycle phases. Nat Methods. 2016;13(12):993-996. doi:10.1038/nmeth.4045

Minor points:
1. The first sentence (line 38): “In normal cells, cell cycle is strictly regulated and organized and comprises in phases G0-G1-S-M.” G2 phase is missed.
2. Lines 45-46: “… the confluent cells in culture or plastic or glass.” Perhaps it should be “… the confluent cells in culture on plastic or glass.”
3. Lines 46-47: “The normal cells such as foreskin fibroblasts cease proliferation and are in contrast inhibited. In contrast, the cancer cells continue to proliferate …”. The first “in contrast” is superfluous, please remove: “The normal cells such as foreskin fibroblasts cease proliferation and are inhibited. In contrast, the cancer cells continue to proliferate …”
4. Incomplete sentence (line 55): “Recently, cancer stem cells (CSCs) are chemo-resistant (17, 18).”
5. Grammatically incorrect sentence (line 58): “For examples, especially FUCCI-transgenic mice showed a distinctly regulated and organized cell-cycle distribution”.
6. The wording “using with” (lines 164, 187, 192) looks excessive, I would write either “using” or “with”.
7. Line 199: “fetal serum bovine”. Please change to the standard “fetal bovine serum”.
8. Lines 214-218: “We showed that oncolytic adenovirus decoyed and trapped quiescent cancer stem cells from quiescent G0/G1 phase to in early S phase, where the decoyed cancer cells became sensitive to chemotherapy (110) (Figure 2B). We also demonstrated that oncolytic adenovirus decoyed and trapped quiescent established tumors from quiescent G0/G1 phase to in early S phase, where the decoyed tumors became sensitive to chemotherapy (110) (Figure 2B).”
These two sentences are mostly identical, please combine they into one sentence, e.g. “We showed that oncolytic adenovirus decoyed and trapped both quiescent cancer stem cells and quiescent established tumors from G0/G1 phase to in early S phase, where the decoyed cancer cells became sensitive to chemotherapy (110) (Figure 2B).”
9. There is no explanation for abbreviations LSC and CML (section 8).

Author Response

Point-by-Point Response to the Reviewer’s Comments

Yano et al.: “Real-time fluorescence image guided cell-cycle perturbation for effective cancer treatment (MS No. Cancers 862286)

Response;

The authors are grateful for the very helpful comments by the reviewer that enabled us to significantly improve the manuscript.

Reviewer #2

Summary

The review by Yano et al. considers a problem of different sensitivity of cancer cells to various treatments depending on cell cycle phases. The authors published a lot of original articles and are world recognized experts in this area. A new concept of “Decoy, trap, and kill cancer therapy” is formulated and justified. This review will be of interest for a wide auditorium of researchers and clinicians. However, some important issues should be corrected before publication.

Major point 1.

Figure 1 shows wrong colors of FUCCI depending of cell cycle phases. In fact, in the original FUCCI, green fluorescence corresponds to S-G2-M (not G2-M as depicted here), and yellow fluorescence corresponds to a short period at the G1/S transition. See Ref. 23, Fig. 1B. Later in the manuscript, there is a correct description (lines 121-123): “FUCCI-red indicates quiescent G0/G1 phase (23), FUCCI-green indicates proliferating late S/G2/M phase, and FUCCI-yellow means early S phase (Figure 1A).” Also, Fig. 2 should be corrected accordingly as it contains explanatory sketch at the bottom where yellow circle means “Cells in S phase” and green circle means “Cells in G2/M phase”.

Response:

These corrections have been made in the revised manuscript.

Major point 2.

In the section 3 (“Fluorescence ubiquitination cell cycle indicator (FUCCI) …”), it is worth noting the variant Fucci4, which resolves all cell cycle phases by using four orthogonal fluorescent indicators: Bajar BT, Lam AJ, Badiee RK, et al. Fluorescent indicators for simultaneous reporting of all four cell cycle phases. Nat Methods. 2016;13(12):993-996. doi:10.1038/nmeth.4045

Response:

FUCCI4 is described in the section 1 in the revised manuscript, and the Bajar et. al. reference is added as #13.

Minor Comment 1.

Line 38: “In normal cells, cell cycle is strictly regulated and organized and comprises in phases G0-G1-S-M.” G2 phase is missed.

Response:

This correction has been made in the revised manuscript.

Minor Comment 2.

Line 45-46: “… the confluent cells in culture or plastic or glass.” Perhaps it should be “… the confluent cells in culture on plastic or glass.”

Response:

In the revised focused manuscript, this point is no longer discussed.

Minor Comment 3.

Line 46-47: “The normal cells such as foreskin fibroblasts cease proliferation and are in contrast inhibited. In contrast, the cancer cells continue to proliferate …”. The first “in contrast” is superfluous, please remove: “The normal cells such as foreskin fibroblasts cease proliferation and are inhibited. In contrast, the cancer cells continue to proliferate …

Response:

In the revised focused manuscript, this point is no longer discussed.

Minor Comment 4.

Line 55: Incomplete sentence (line 55): “Recently, cancer stem cells (CSCs) are chemo-resistant (17, 18).

Response:

This correction has been made in the revised manuscript.

Minor Comment 5.

Line 58: “For examples, especially FUCCI-transgenic mice showed a distinctly regulated and organized cell-cycle distribution”.

Response:

In the revised focused manuscript, this sentence has been eliminated.

Minor Comment 6.

Lines 164, 187, 192: “The wording “using with” (lines 164, 187, 192) looks excessive, I would write either “using” or “with”

Response:

These corrections have been made in the revised manuscript.

Minor Comment 7.

Line 199: “fetal serum bovine”. Please change to the standard “fetal bovine serum”.

Response:

This correction have been made in the revised manuscript.

Minor Comment 8.

Lines 214-218: “We showed that oncolytic adenovirus decoyed and trapped quiescent cancer stem cells from quiescent G0/G1 phase to in early S phase, where the decoyed cancer cells became sensitive to chemotherapy (110) (Figure 2B). We also demonstrated that oncolytic adenovirus decoyed and trapped quiescent established tumors from quiescent G0/G1 phase to in early S phase, where the decoyed tumors became sensitive to chemotherapy (110) (Figure 2B).” 
These two sentences are mostly identical, please combine they into one sentence, e.g. “We showed that oncolytic adenovirus decoyed and trapped both quiescent cancer stem cells and quiescent established tumors from G0/G1 phase to in early S phase, where the decoyed cancer cells became sensitive to chemotherapy (110) (Figure 2B).”

Response:

In the revised manuscript, these sentences have been combined.

Minor Comment 9.

There is no explanation for abbreviations LSC and CML (section 8).

Response:

In the revised focused manuscript, LSC and CML are no longer discussed.

Reviewer 3 Report

The title of this review, "Real-time fluorescence image guided cell-cycle perturbation for effective cancer treatment" is intriguing, but misleading. Exposing cells to an anti-cancer drug is not "cancer treatment". Even before, and with the advent of cell cycle reporter systems, researchers have been exposing cells to different drugs to study cell cycle specific effects, so this idea is not new. I agree that the approach has seen a renaissance with live imaging approaches, and tumors grown in animal models, but this is not articulated well in the review. The review is very difficult to read, in part due to significant English writing issues, but also because of organization, choice of terms, and oversimplification of the cancer problem. The authors chose to focus in the FUCCI system for tracking cell cycle, however other fluorescent systems exist (e.g. Cdk sensors) that should be included as this is a review article.

Specific points:

  1. The instances of English writing problems are numerous. Rather than list them, the services of a professional or native English speaker must be used.
  2. With regard to the discussion of "anti-mitotics", taxanes have very complicated pharmacology, with different cell phenotypes at different, specific concentrations. There is also evidence from Weaver, Mitchison, and others that powerful anti-tumor effects after taxanes are not due to effects on mitosis itself. The current discussion is superficial and needs to be updated.
  3. Example of writing issue, line 51-52, "However, most cytotoxic agents have limited for for solid cancers."
  4. A comment on cancer stem cells being resistant to chemotherapy is made, it states they are resistant because they are quiescent, this is true, but not the only reason, in a review it cannot be this superficial. Even growing cancer stem cells show less cell death. Additional reasons need to be discussed.
  5. The phrase "cell cycle decoyers" is confusing, "decoyers" isn't a word. Perhaps simply "decoys" can be used?
  6. The idea of targeting quiescent cells, or highly-related senescent cells, is an exciting area of drug development. This section needs to be further developed.
  7.  

Author Response

Point-by-Point Response to the Reviewer’s Comments

Yano et al.: “Real-time fluorescence image guided cell-cycle perturbation for effective cancer treatment (MS No. Cancers 862286)

Response;

The authors are grateful for the very helpful comments by the reviewer that enabled us to significantly improve the manuscript.

Reviewer #3:

The title of this review, "Real-time fluorescence image guided cell-cycle perturbation for effective cancer treatment" is intriguing, but misleading. Exposing cells to an anti-cancer drug is not "cancer treatment". Even before, and with the advent of cell cycle reporter systems, researchers have been exposing cells to different drugs to study cell cycle specific effects, so this idea is not new. I agree that the approach has seen a renaissance with live imaging approaches, and tumors grown in animal models, but this is not articulated well in the review. The review is very difficult to read, in part due to significant English writing issues, but also because of organization, choice of terms, and oversimplification of the cancer problem. The authors chose to focus in the FUCCI system for tracking cell cycle, however other fluorescent systems exist (e.g. Cdk sensors) that should be included as this is a review article.

Response:

The title of the revised manuscript has been changed. The revised manuscript focuses on how the revolutionary aspect of FUCCI, especially that it uniquely images cell-cycle dynamics in real time in vivo as well as in vitro are emphasized. Mistakes have been corrected and English has been improved.

Comment 1.

The instances of English writing problems are numerous. Rather than list them, the services of a professional or native English speaker must be used.

Response:

Co-author, Dr. Hoffman, a native English speaker, has improved the English.

Comment 2.

With regard to the discussion of "anti-mitotics", taxanes have very complicated pharmacology, with different cell phenotypes at different, specific concentrations. There is also evidence from Weaver, Mitchison, and others that powerful anti-tumor effects after taxanes are not due to effects on mitosis itself. The current discussion is superficial and needs to be updated.

Response:

The revised focused manuscript no longer discusses these points.

Comment 3.

Example of writing issue, line 51-52, "However, most cytotoxic agents have limited for for solid cancers."

Response:

Sentences such as this have been corrected in the revised manuscript

Comment 4.

A comment on cancer stem cells being resistant to chemotherapy is made, it states they are resistant because they are quiescent, this is true, but not the only reason, in a review it cannot be this superficial. Even growing cancer stem cells show less cell death. Additional reasons need to be discussed.

Response:

The revised manuscript states that quiescence is just one reason cancer stem cells may be chemoresistant. Further discussion of stem cell biology is not in the scope of the revised focused review.

Comment 5.

The phrase "cell cycle decoyers" is confusing, "decoyers" isn't a word. Perhaps simply "decoys" can be used?

Response:

This correction has been made in the revised manuscript.

Comment 6.

The idea of targeting quiescent cells, or highly-related senescent cells, is an exciting area of drug development. This section needs to be further developed.

Response:

The idea of targeting quiescent stem cells using FUCCI to visualize their cell-cycle dynamics in real time is expanded in the revised manuscript and it is emphasized that quiescent cancer stem cells can be very dangerous to the host as they commence re-cycling after cessation of chemotherapy and may become more malignant. Senescence is not discussed as FUCCI cannot yet identify senescent cells.

Round 2

Reviewer 1 Report

The revised manuscript is significantly improved over the first draft, but it is still heavily focused on the authors own work. It blurs the line in places between being an unbiased review or simply an advertisement for previously published papers from the lab, especially with the large amounts of published data and non-published results, which aren’t really appropriate for a review article. 

There are still many instances of mis-leading statements and overreaching conclusions which should be addressed. Many of these still lack references to support the claims. The language has been improved but could do with further refinement across the whole article.

Some examples are listed below and should be used as a guide and not an exhaustive list:

Line 28: Not true, there are other ways to extract cell cycle information, for example, tagged PCNA, tagged cyclins, CDK and other kinase FRET-biosensors and even NLS-tagged GFP can be used to determine various cell cycle phases and hence monitor different aspects of cell cycle status. Hence, this statement needs to be softened. Furthermore, some mention and or comparison to these other methods would be beneficial. Essentially, the review should provide some reason as to why and how FUCCI is better (or worse) and why it should (or shouldn’t) be used.

Line 37: This is a bit clumsy and poorly worded. You state 5-stages here, but then state only 4 for cancer cells, which is undermined by next sentence where you state that cancer cells are often in G0. Suggest you change to 4-active phases (G1, S, G2 and M) with inactive/quiescent G0 phase. You can then state that the regulation of these phases is commonly disrupted in and is a hallmark of cancer cells.

Line 81: This is poorly written and unclear. The problem with the standard two-plasmid system is that they expression is not linked and can vary widely. For example, you can have a cell that massively over-expresses the Red plasmid and weakly expresses the green. This can flood the colour balance, making it difficult to see the yellow colour change. Furthermore, too high expression of red, can prevent total destruction, leading to red remaining throughout the entire cell cycle. Similarly, cells may only express one of the plasmids, again biasing results. Further, while figure 1 provides an overview of standard FUCCI, the review is lacking a figure showing the different types of FUCCI systems and their advantages and disadvantages. Such a figure would be far more appropriate than the much of the re-hashed data shown in figures shown in 2-9.

Line 101: Delete ‘real’.

Line 110: delete basically.

Line 132: ‘G2 DNA checkpoint’ assume you mean DNA damage…although FUCCI2 by itself cannot conclusively state that cells are in G2 and hence this is also misleading, cells could be in S phase. By our own admission, you would need to use FUCCI-CA

Line 157: This section randomly starts with (33, 34).

Line 164: should read: with regard to ‘the’, not ‘then’ degree.

Section 6-9: These sections are highly/entirely focused on only the authors previous work. These should be more balanced. There is also no clear point to what these sections are ‘reviewing’, it’s very much just a list of statements, there is also no common link between these and hence the review is disjointed.

Figure 2-9: Contains published results pulled from a previous publications and/or unpublished data, which are not really appropriate for a review. For example, Figure 4 is nearly identical to that previously published in by the authors in Figure 2 of

Yano S, Hoffman RM. Cells. 2018;7(10):168. Published 2018 Oct 14. doi:10.3390/cells7100168.

Section 9-10: Similar to sections 6-9 are not well introduced, and there is no clear link to between them and the rest of the review. There needs to be a clearer logical flow of the review. Essentially, the review has no clear point, other than to promote the various previous findings of the lab, and hence lacks a clear direction and focus. Without this focus, it drifts into being a collection of statements, that are loosely collated around the use of FUCCI.

Author Response

Point-by-Point Response to the Reviewer’s Comments

Yano et al.: “Real-time fluorescence image guided cell-cycle perturbation for effective cancer treatment (MS No. Cancers 862286)

Response;

The authors are grateful for the very helpful comments by the reviewer that enabled us to significantly improve the manuscript.

Reviewer #1

Summary

The revised manuscript is significantly improved over the first draft, but it is still heavily focused on the authors own work. It blurs the line in places between being an unbiased review or simply an advertisement for previously published papers from the lab, especially with the large amounts of published data and non-published results, which aren’t really appropriate for a review article. 

There are still many instances of mis-leading statements and overreaching conclusions which should be addressed. Many of these still lack references to support the claims. The language has been improved but could do with further refinement across the whole article.

Response:

The revised review is more balanced. The previous-published images used in this manuscript are essential for making discussion clear. The unpublished images make the review more timely. All published work is fully referenced. Our laboratory is the pioneer for much of the use of FUCCI images of individual cancer cells within solid tumors and in 3D culture. There is some emphasis on our work, but we have cited much other work. In the revised manuscript we have tried to review all FUCCI work in this area and be more balanced and eliminate misleading sentences.

Comment 1;

Line 28: Not true, there are other ways to extract cell cycle information, for example, tagged PCNA, tagged cyclins, CDK and other kinase FRET-biosensors and even NLS-tagged GFP can be used to determine various cell cycle phases and hence monitor different aspects of cell cycle status. Hence, this statement needs to be softened. Furthermore, some mention and or comparison to these other methods would be beneficial. Essentially, the review should provide some reason as to why and how FUCCI is better (or worse) and why it should (or shouldn’t) be used.

Response:

Section 8; We have compared FUCCI with other cell-cycle indicators, and indicated the unique advantages of FUCCI in Section 8.

Comment 2;

Line 37: This is a bit clumsy and poorly worded. You state 5-stages here, but then state only 4 for cancer cells, which is undermined by next sentence where you state that cancer cells are often in G0. Suggest you change to 4-active phases (G1, S, G2 and M) with inactive/quiescent G0 phase. You can then state that the regulation of these phases is commonly disrupted in and is a hallmark of cancer cells.

Response:

Lines 45, 46: This change has been made.

Comment 3;

Line 81: This is poorly written and unclear. The problem with the standard two-plasmid system is that they expression is not linked and can vary widely. For example, you can have a cell that massively over-expresses the Red plasmid and weakly expresses the green. This can flood the colour balance, making it difficult to see the yellow colour change. Furthermore, too high expression of red, can prevent total destruction, leading to red remaining throughout the entire cell cycle. Similarly, cells may only express one of the plasmids, again biasing results. Further, while figure 1 provides an overview of standard FUCCI, the review is lacking a figure showing the different types of FUCCI systems and their advantages and disadvantages. Such a figure would be far more appropriate than the much of the re-hashed data shown in figures shown in 2-9.

Response:

Lines 85-89: We have now clearly stated the problems with the original FUCCI and how these problems have been overcome by modern versions of FUCCI.

Comment 4;

Line 101: Delete ‘real’.

Response:

Line 107: ‘Real’ has been deleted.

Comment 5;

Line 110: delete basically.

Response:

Line 116: ‘Basically’ has been deleted.

Comment 6;

Line 132: ‘G2 DNA checkpoint’ assume you mean DNA damage…although FUCCI2 by itself cannot conclusively state that cells are in G2 and hence this is also misleading, cells could be in S phase. By our own admission, you would need to use FUCCI-CA

Response:

Lines 145: We have eliminated DNA check point.

Comment 7;

Line 157: This section randomly starts with (33, 34).

Response:

This has been corrected.

Comment 8;

Line 164: should read: with regard to ‘the’, not ‘then’ degree.

Response:

Line 162: This has been corrected.

Comment 9;

Section 6-9: These sections are highly/entirely focused on only the authors previous work. These should be more balanced. There is also no clear point to what these sections are ‘reviewing’, it’s very much just a list of statements, there is also no common link between these and hence the review is disjointed.

Response:

Former sections 6-9 have been re-organized in to one section that is focused on use of FUCCI to help to solve the recalcitrant problem of quiescent cancer cells within tumors.

Comment 10;

Figure 2-9: Contains published results pulled from a previous publications and/or unpublished data, which are not really appropriate for a review. For example, Figure 4 is nearly identical to that previously published in by the authors in Figure 2 of

Yano S, Hoffman RM. Cells. 2018;7(10):168. Published 2018 Oct 14. doi:10.3390/cells7100168.

Response:

The present paper is a review and the author respectfully state the benefit of using previously published figures to make the discussion clear and attractive. We have fully referenced all previously published figures and will obtain permission from the original journal to use each of them. To make the review timely, we have added some unpublished data that also clarify our discussion. FUCCI is all about images and that is why the authors respectively emphasize the importance of including the images we have chosen.

Comment 11;

Section 9-10: Similar to sections 6-9 are not well introduced, and there is no clear link to between them and the rest of the review. There needs to be a clearer logical flow of the review. Essentially, the review has no clear point, other than to promote the various previous findings of the lab, and hence lacks a clear direction and focus. Without this focus, it drifts into being a collection of statements, that are loosely collated around the use of FUCCI.

Response:

The review is now much more focused with a main mission of using FUCCI to develop new strategies to target quiescent cancer cells, which by their very quiescence are resistant to cytotoxic chemotherapy and are currently a recalcitrant problem for treating solid cancers.

Reviewer 2 Report

Authors revised the manuscript very significantly. Now I have only one minor remark:

Lines 60-64: “FUCCI utilizes genes such as Cdt1 and Geminin linked to different-color fluorescent reporters that are only expressed in specific phases of the cell cycle. FUCCI comprises two plasmids; Cdt1-mKO2 (orange florescent protein) expressed only in G1 62 phase, Geminin-mAG (green florescent protein) expressed in late-S/G2/M phases, and FUCCI-yellow expressed in early S phase where the expression of Cdt1-mKO2 was same as that of Geminin-mAG.”

The term “expressed” is not fully correct for description of FUCCI, as it is commonly used for “gene expression”. Meantime, FUCCI-coding genes are always actively expressed, but corresponding protein products are stable in some cell cycle phases and unstable in others. 

Author Response

Point-by-Point Response to the Reviewer’s Comments

Yano et al.: “Real-time fluorescence image guided cell-cycle perturbation for effective cancer treatment (MS No. Cancers 862286)

Response;

The authors are grateful for the very helpful comments by the reviewer that enabled us to significantly improve the manuscript.

Reviewer #2

Authors revised the manuscript very significantly. Now I have only one minor remark: Lines 60-64: “FUCCI utilizes genes such as Cdt1 and Geminin linked to different-color fluorescent reporters that are only expressed in specific phases of the cell cycle. FUCCI comprises two plasmids; Cdt1-mKO2 (orange florescent protein) expressed only in G1 62 phase, Geminin-mAG (green florescent protein) expressed in late-S/G2/M phases, and FUCCI-yellow expressed in early S phase where the expression of Cdt1-mKO2 was same as that of Geminin-mAG.”

The term “expressed” is not fully correct for description of FUCCI, as it is commonly used for “gene expression”. Meantime, FUCCI-coding genes are always actively expressed, but corresponding protein products are stable in some cell cycle phases and unstable in others. 

Response:

Lines 69-73; We have corrected them.

Round 3

Reviewer 1 Report

No comments.